# Structural insights into the molecular mechanisms of myasthenia gravis and their therapeutic implications

Kaori Noridomi[1], Go Watanabe[2], Melissa N Hansen[3], Gye Won Han[4], Lin Chen[1,2,3]*

[1]Department of Chemistry, University of Southern California, Los Angeles, United States; [2]USC Norris Comprehensive Cancer Center, Keck School of Medicine, University of Southern California, Los Angeles, United States; [3]Molecular and Computational Biology, Department of Biological Sciences, University of Southern California, Los Angeles, United States; [4]Department of Chemistry, Bridge Institute, University of Southern California, Los Angeles, United States

**Abstract** The nicotinic acetylcholine receptor (nAChR) is a major target of autoantibodies in myasthenia gravis (MG), an autoimmune disease that causes neuromuscular transmission dysfunction. Despite decades of research, the molecular mechanisms underlying MG have not been fully elucidated. Here, we present the crystal structure of the nAChR $\alpha$1 subunit bound by the Fab fragment of mAb35, a reference monoclonal antibody that causes experimental MG and competes with ~65% of antibodies from MG patients. Our structures reveal for the first time the detailed molecular interactions between MG antibodies and a core region on nAChR $\alpha$1. These structures suggest a major nAChR-binding mechanism shared by a large number of MG antibodies and the possibility to treat MG by blocking this binding mechanism. Structure-based modeling also provides insights into antibody-mediated nAChR cross-linking known to cause receptor degradation. Our studies establish a structural basis for further mechanistic studies and therapeutic development of MG.

*For correspondence: linchen@usc.edu

## Introduction

The nicotinic acetylcholine receptor (nAChR) at the neuromuscular junction (NMJ) is a ligand-gated ion channel that mediates rapid signal communication between spinal motor neurons and the muscle cells. This receptor is also a major target of autoimmune antibodies in patients with myasthenia gravis (MG), an autoimmune disease that afflicts more than 20 in 100,000 people (*Lindstrom, 2000*; *Vincent et al., 2001*). MG is the first and, so far, only autoimmune disease with well-defined autoantigen target; the binding of nAChR by MG antibodies leads to complement-mediated lysis of the postsynaptic structure and internalization of the receptor, thereby disrupting neuromuscular transmission (*Engel and Arahata, 1987*; *Drachman et al., 1978*; *Gomez et al., 2010*). The majority of MG cases can be diagnosed by the detection of autoantibodies to human muscle nAChR, and current treatment options include the use of acetylcholine esterase inhibitors, non-specific immunosuppressive drugs, plasmapheresis and thymectomy. Most of these treatments are for symptomatic control except for thymectomy that may lead to disease remission. Other therapeutic approaches to treating MG, such as nAChR-specific immunosuppressive therapy (*Luo and Lindstrom, 2015*), need to be explored.

The autoimmune nature of MG was first suggested by the discovery of experimental autoimmune myasthenia gravis (EAMG) induced in rabbits immunized with nAChR purified from *Electrophorus*

**eLife digest** Myasthenia gravis is a disease that causes chronic weakness in muscles. It affects more than 20 in every 100,000 people and diagnosis is becoming more common due to increased awareness of the disease. However, most current treatments only temporarily relieve symptoms so there is a need to develop more effective therapies.

The disease occurs when the immune system produces molecules called antibodies that bind to and destroy a receptor protein called nAChR. This receptor is normally found at the junctions between nerve cells and muscle cells, and its destruction disrupts communication between the nervous system and the muscle. However, it is not known exactly how these antibodies bind to nAChR, partly due to the lack of a detailed three-dimensional structure of the antibodies and nAChR together.

The human nAChR protein is made up of several subunits, including one called alpha1 that is the primary target of Myasthenia gravis antibodies. Noridomi et al. used a technique known as X-ray crystallography to generate a highly detailed three-dimensional model of the structure of the alpha1 subunit with an antibody from rats that acts as in a similar way to human Myasthenia gravis antibodies. The structure reveals the points of contact between the antibodies and a core region of the nAChR alpha1 subunit and suggests that many different Myasthenia gravis antibodies may bind to nAChR in the same way.

These findings may aid the development of drugs that bind to and disable Myasthenia gravis antibodies to relieve the symptoms of the disease.

*electricus* (*Patrick and Lindstrom, 1973*). Subsequent studies with passive transfer of MG patient serum or purified nAChR antibodies to induce EAMG further established nAChR antibodies as the major pathological agents of MG (*Toyka et al., 1975*; *Lindstrom et al., 1976*). In fact, more than 85% of MG patients carry nAChR antibodies (*Lindstrom, 2000*; *Vincent et al., 2001*; *Meriggioli and Sanders, 2009*). However, the total amount of nAChR antibodies in the serum of MG patients does not seem to correlate with disease severity, suggesting that various nAChR antibodies that bind different regions on nAChR may contribute differently to this disease (*Somnier, 1993*; *Berrih-Aknin, 1995*; *Mossman et al., 1988*; *Tzartos et al., 1998*).

Mammalian muscle nAChR has a pentameric structure composed of two $\alpha$1, one $\beta$1, one $\delta$, and one $\varepsilon$ (adult form) or $\gamma$ (fetal form) subunit(s) (*Unwin, 2005*). Extensive studies suggest that antibodies to $\alpha$1 play a major role in MG pathology (*Sideris et al., 2007*; *Tzartos et al., 2008*, *1987*; *Kordas et al., 2014*). Furthermore, more than half of all autoantibodies in MG and EAMG bind an overlapping region on the nAChR $\alpha$1 subunit, known as the main immunogenic region (MIR) (*Tzartos et al., 1998*). The MIR is defined by the ability of a single rat monoclonal antibody (mAb), mAb35, to inhibit the binding of about 65% autoantibodies from MG patients or rats with EAMG (*Tzartos and Lindstrom, 1980*; *Tzartos et al., 1982*, *1983*). Subsequent studies have mapped MIR to a peptide region that spans residues 67–76 on nAChR $\alpha$1 (*Barkas et al., 1988*; *Tzartos et al., 1988*). Monoclonal antibodies directed to the MIR can passively transfer EAMG and possess all the key pathological functions of serum autoantibodies from MG patients (*Tzartos et al., 1987*). Moreover, a recent study showed that titer levels of MIR-specific antibody from MG patients, rather than the total amount of nAChR antibodies, correlate with disease severity (*Masuda et al., 2012*). These observations suggest that antibodies binding to the MIR on nAChR $\alpha$1 play a major role in the pathogenesis of MG (*Tzartos et al., 1998*).

The myasthenogenic role of nAChR was established more than four decades ago. Since then, extensive efforts have been put into characterizing the interactions between MG antibodies and nAChR using biochemical (*Barkas et al., 1988*; *Tzartos et al., 1988*; *Das and Lindstrom, 1989*; *Saedi et al., 1990*; *Papadouli et al., 1990*, *1993*; *Luo et al., 2009*; *Morell et al., 2014*), structural (*Dellisanti et al., 2007a*; *Beroukhim and Unwin, 1995*; *Kontou et al., 2000*; *Poulas et al., 2001*), and modeling approaches (*Kleinjung et al., 2000*). These studies aimed to understand the basic mechanisms of MG and also the structure/function of nAChR in order to develop effective diagnosis and treatment for MG. However, exactly how antibodies bind and functionally affect nAChR has not

been fully elucidated since no high-resolution structure of the complex between MG antibodies and nAChR was available. Here we describe the first crystal structure of muscle nAChR α1 subunit bound by an EAMG antibody at 2.61 Å resolution and present detailed analyses of the molecular interactions in myasthenia gravis. These structural analyses, in the context of the large amount of biochemical and functional data from previous MG research, provide unprecedented insights into the molecular mechanisms of MG and a basis for developing more effective diagnosis and treatment for this debilitating disease.

## Results

### Crystal structures of the antibody/receptor complexes

mAb35 was chosen for structural analysis because it shares many functional characteristics with serum antibodies from MG patients and has been used as a reference MG antibody in extensive biochemical and functional studies (*Tzartos et al., 1998*, *1981*). Although mAb35 is derived from rat immunized with *Electrophorus* AChR, it competes with more than two thirds of serum antibodies from MG patients (*Tzartos et al., 1982*). At the functional level, mAb35 binds complement causing focal lysis of the postsynaptic membrane, cross-links AChRs thereby increasing their internalization, and can passively transfer EAMG (*Tzartos et al., 1987*). To facilitate crystallization, we used the Fab fragment of mAb35 (Fab35) and also included α-bungarotoxin (α-Btx) to stabilize flexible regions of nAChR α1 ECD that may hinder crystallization. We used a mutant of nAChR α1 ECD that contains three stabilization mutations, referred to as α211 as described previously (*Dellisanti et al., 2007a*). Although mAb35 does not bind well to the denatured receptor, it has been shown to bind natively folded nAChR α1 with high affinity ($K_d$ =~2 nM) (*Luo et al., 2009*). Our native PAGE analysis showed that nAChR α1 ECD, α-Btx and Fab35 can form a well-defined ternary complex (*Figure 1a*), and the ternary complex of Fab35/nAChR α1 ECD/α-Btx was stable during purification by gel filtration. We obtained ternary complex crystals with both human and mouse nAChR α1 ECD and solved the structures (2.61 Å and 2.70 Å resolution, respectively) by molecular replacement (*Supplementary file 1*). A detailed comparison between the two complex structures is presented in *Figure 1—figure supplements 1* and *2*. Here, we focus our structural description and analysis on the antibody/receptor interface, which appears identical between the human and mouse complexes.

The Fab35 binds to nAChR α1 in an upright orientation, away from the α-Btx (*Figure 1b and c*). The Fab35 binding sites on nAChR α1 include the MIR and the N-terminal helix; the buried solvent accessible area of the complex is 899 Å$^2$. Fab35 has the canonical IgG antibody structure where the complementarity determining regions (CDRs) from the heavy chain, CDR-H2 and CDR-H3, and the light chain, CDR-L3, form the binding site of nAChR α1 (*Figure 1d*). The interface of their interaction is characterized by mutual insertion of loops into pockets of binding partners. On the receptor side (*Figure 2a*), the MIR loop inserts deeply into a surface pocket between the variable domains of the heavy and light chains ($V_H$ and $V_L$), whereas the N-terminal helix sits into a groove on the surface of the heavy chain. On the Fab35 side (*Figure 2b*), the CDR-H3 inserts into a surface pocket formed by the N-terminal helix, the loop following the N-terminal helix, the MIR and the loop preceding the MIR.

Superposition of the structures of mouse nAChR α1 ECD in the ternary Fab35/nAChR α1 ECD/α-Btx complex with that in the binary nAChR α1 ECD/α-Btx complex (PDB ID, 2QC1) (*Dellisanti et al., 2007a*) shows that the nAChR α1 structure remains the same in the two complex states (*Figure 1—figure supplement 3a*). Moreover, the orientation of the side chain of most nAChR α1 residues involved in Fab35 binding (see below) is similar between the two complexes (*Figure 1—figure supplement 3b*). This structural comparison suggests that Fab35 recognizes and binds a well-defined and preformed conformation of the nAChR α1 ECD.

### Detailed interactions at the binding interface

Residues from Fab35 and nAChR α1 that are within 4.5 Å from each other at the binding interface were mapped as contacting residues. As shown in *Supplementary file 2*, Fab35-binding residues on nAChR α1 are mostly located on the MIR loop (highlighted in light green in the table) and the N-terminal helix (highlighted in yellow in the table). While the MIR loop extensively interacts with residues from both the heavy and light chains of Fab35, the N-terminal helix interacts exclusively with

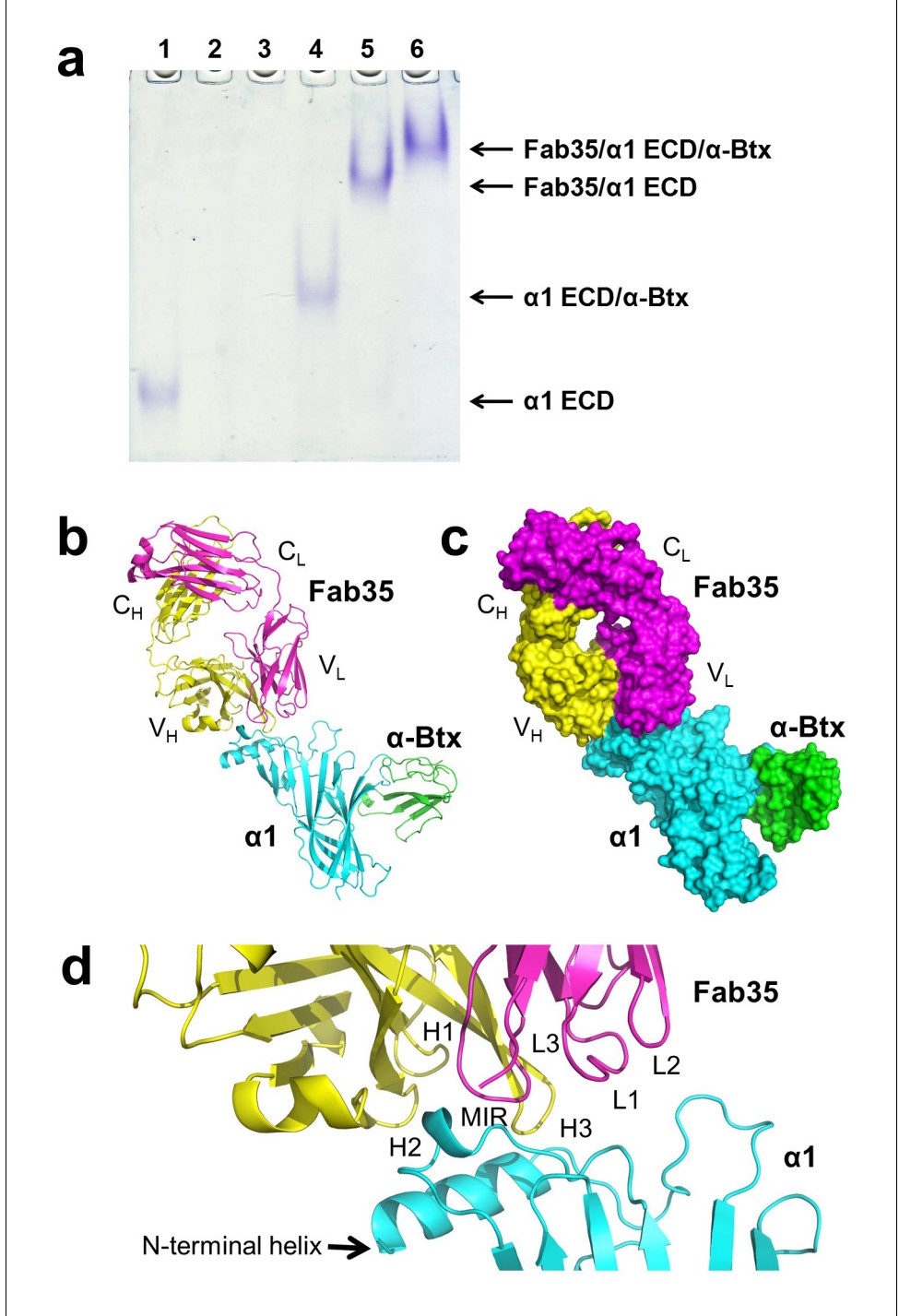

**Figure 1.** The ternary complex of nAChR α1 ECD bound by Fab35 and α-Btx. (a) Gel shift assay. Native PAGE showed the formation of the ternary complex of nAChR α1 ECD, α-Btx and Fab35. Lane 1: nAChR α1 ECD alone (labeled as α1 ECD), Lane 2: α-Btx alone, Lane 3: Fab35 alone, Lane 4: nAChR α1 ECD plus α-Btx, Lane 5: nAChR α1 ECD plus Fab35, and Lane 6: nAChR α1 ECD plus α-Btx plus Fab35. Note that α-Btx in Lane 2 and Fab35 in Lane 3 were not visible because both proteins migrated upward due to their net positive charges under the experimental condition. (b) Ribbon representation of nAChR α1 ECD (α1: cyan) bound by α-Btx (green) and Fab35 (heavy chain, H: yellow and light chain, L: magenta). The variable domains ($V_H$ and $V_L$) and the constant domains ($C_H$ and $C_L$) of Fab35 are indicated accordingly. This color scheme is kept the same throughout illustration unless noted otherwise. (c) Surface representation of the ternary complex. (d) Zoomed-in view of the binding interface. The complementarity determining regions (CDRs) of the heavy chain (CDR-H1, CDR-H2, and CDR-H3) are

*Figure 1 continued on next page*

*Figure 1 continued*

indicated as H1, H2, and H3, respectively. Those of the light chain (CDR-L1, CDR-L2 and CDR-L3) are indicated as, L1, L2, and L3, respectively.

The following figure supplements are available for figure 1:

**Figure supplement 1.** Key structural features of the human nAChR α1 ECD.

**Figure supplement 2.** Structural differences between the human and mouse nAChR α1 ECDs.

**Figure supplement 3.** Structural comparison of mouse nAChR α1 ECDs in the ternary complex of Fab35/nAChR α1 ECD/α-Btx and the binary complex of nAChR α1 ECD/α-Btx.

residues from the heavy chain. The contacting analysis also revealed several residues on nAChR α1 that make numerous contacts to Fab35. Four such 'hotspots' of binding were identified: Asn68 and Asp71 from the MIR loop and Arg6 and Lys10 from the N-terminal helix. As described below, each of these four 'hotspots' anchors an extensive network of interactions that display remarkable chemical complementarities (*Figure 3*). The interface interactions are well defined by electron densities (*Figure 3—figure supplement 1a–e*).

Asp71 forms a salt bridge with Arg50 of $V_H$ and a hydrogen bond with Tyr95 of $V_L$ (*Figure 3a*). Asp71 also forms hydrogen bonds with two interfacial water molecules, $H_2O$ 4 and $H_2O$ 5. $H_2O$ 4 in turn forms hydrogen bonds to the main chain amide of Asn68 of the α1 and the main chain carbonyl group of Ala103 of $V_H$. $H_2O$ 5 in turn forms hydrogen bonds to the main chain carbonyl group of Tyr91 of $V_L$ and the main chain amide of Asn105 of $V_H$. The adjacent Tyr72 can be considered as part of the Asp71 'hotspot': Tyr72 not only mediates the packing interactions between the MIR loop and the N-terminal helix but also makes extensive contacts to the antibody, including hydrogen

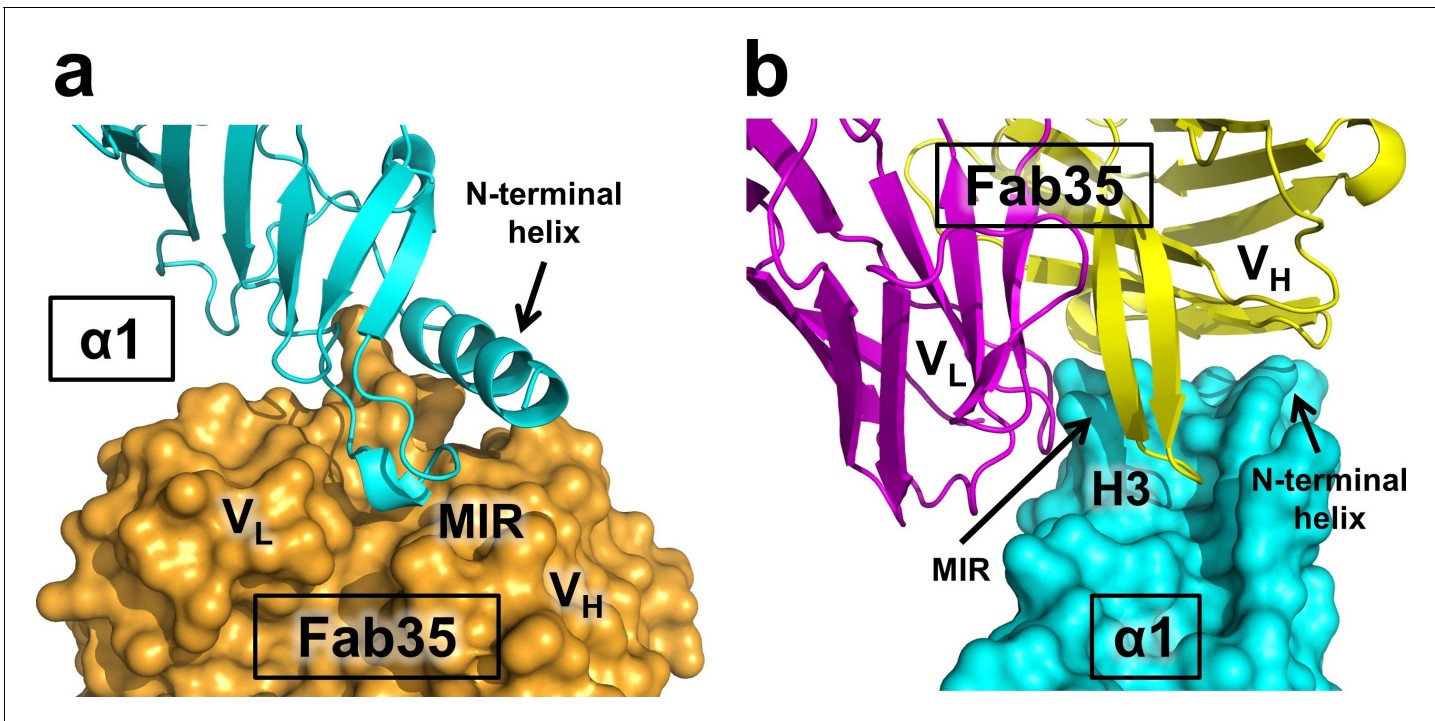

**Figure 2.** Mutual insertion of loops into pockets of binding partners. (**a**) The MIR loop of nAChR α1 inserts into a surface pocket between the variable domains of the heavy and light chains ($V_H$ and $V_L$) of Fab35 (orange) while the N-terminal helix sits into a groove on the surface of the heavy chain. (**b**) The CDR-H3 (H3) from the heavy chain of Fab35 inserts into a surface pocket between the MIR loop and the N-terminal helix on the nAChR α1 ECD.

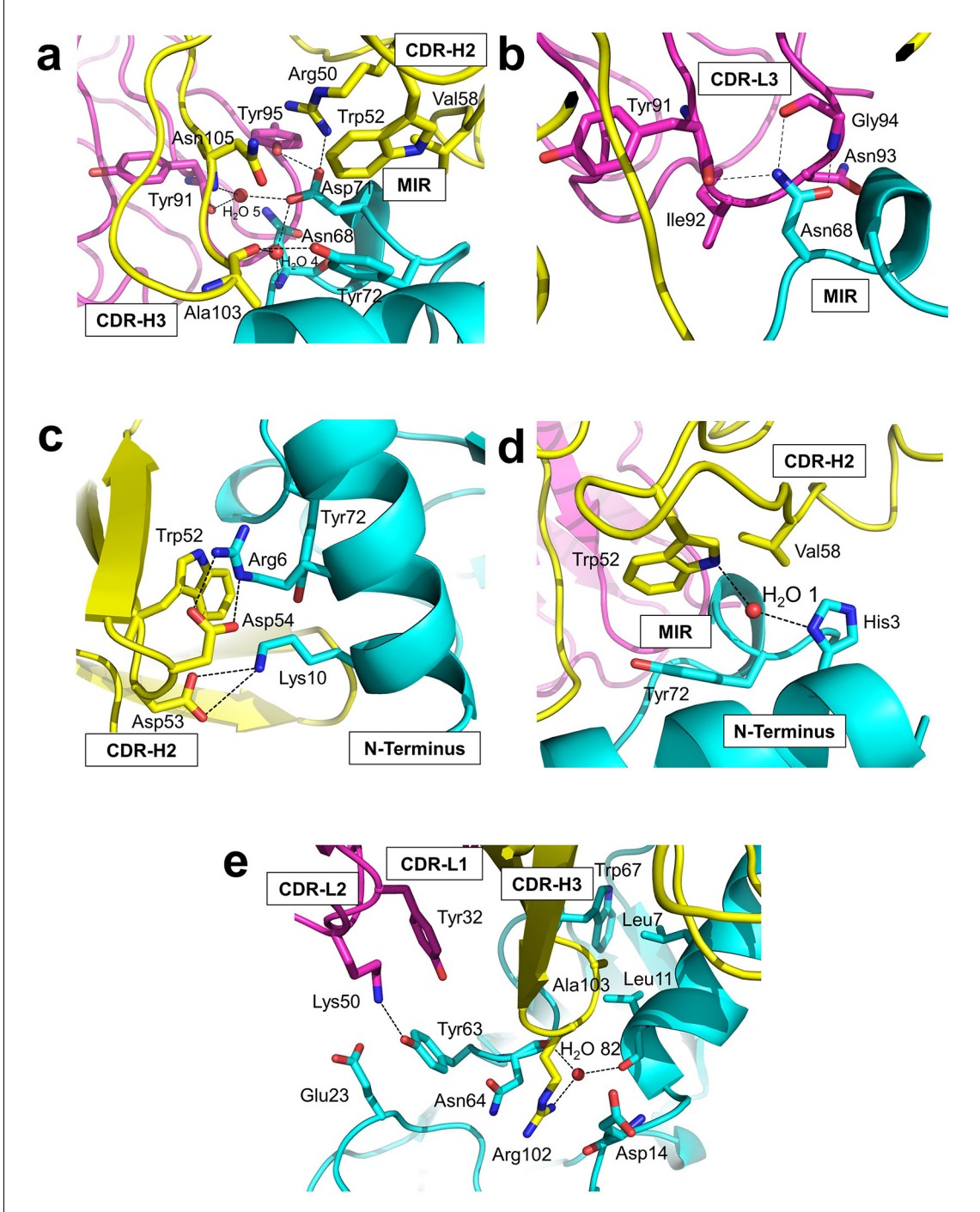

**Figure 3.** Detailed interactions at the interface between Fab35 and nAChR α1 ECD. (a) Binding interactions at the Asp71 site of α1 (located at the MIR). (b) Binding interactions at the Asn68 site of α1 (located at the MIR). (c) Binding interactions surrounding Arg6 and Lys10 of α1 (located at the N-terminus of α1). (d) Binding interactions mediated by His3 of α1 (located at the N-terminus of α1). (e) Binding interactions at the CDR-H3 loop of Fab35. Interacting residues are represented by stick model and are colored according to their protein subunits. Water molecules are represented by red spheres.

The following figure supplement is available for figure 3:

**Figure supplement 1.** $F_o$-$F_c$ omit maps of the interface between Fab35 and nAChR α1 ECD.

bonding to Ala103 and van der Waals contacts to Trp52, Val58, and Asn105 of $V_H$. At Asn68, another hotspot site (*Figure 3b*), the amide group of Asn68 side chain forms three hydrogen bonds with $V_L$ (the carbonyl of Tyr91 main chain and the amide and carbonyl of Gly94 main chain). The side chain of Asn68 also makes van der Waals contact to Ile92 and Asn93 of $V_L$. Both Asn68 and Asp71 together, extending from the tip of the MIR loop, insert deeply into the antigen-binding site of Fab35 and make extensive contacts with Fab35 residues.

The N-terminal helix of nAChR α1 engages in extensive interactions with $V_H$ of Fab35. These interactions are centered at Arg6 and Lys10 residues (*Figure 3c*). The guanidinium head group of Arg6 forms bidendate hydrogen bonds with Asp54 and cation-π stacking with Trp52 of $V_H$. The aliphatic side chain of Arg6 also makes van der Waals contacts to the aromatic ring of Trp52. This interaction network is extended by the nearby nAChR α1 residues, Lys10 and His3 (*Figure 3c and d*, respectively). Lys10 forms salt bridges with Asp53 of $V_H$ (*Figure 3c*); its side chain also makes van der Waals contacts to numerous $V_H$ residues (not shown). His3 makes a water-mediated ($H_2O$ 1) hydrogen bond with Trp52 and van der Waals contact to Val58 of $V_H$ (*Figure 3d*).

An interesting structural feature of the antibody/receptor interface is the insertion of the CDR-H3 loop into a surface pocket on nAChR α1 (*Figure 3e*). The tip of the CDR-H3 loop, including Arg102 and Ala103, makes extensive van der Waals contacts to the surrounding receptor residues. The guanidinium group of Arg102 is sandwiched by the carboxylic amide of Asn64 side chain and the carbonyl of Asp14 main chain of nAChR α1 in a parallel orientation that may favor π-stacking. Arg102 also forms water-mediated ($H_2O$ 82) hydrogen bonds with the main chain carbonyl groups of Leu11 and Asn64 of nAChR α1. Adjacent to the CDR-H3 interaction site, Tyr63 of nAChR α1 forms a hydrogen bond with Lys50 of the CDR-L2, which is stabilized by a cation-π interaction with Tyr32 of the CDR-L1. Lys50 of the CDR-L2 also engages in electrostatic interaction with Glu23 of nAChR α1. These interactions expand the binding interface from the MIR and the N-terminal helix to the loop region between the N-terminal helix and the β-strand β1 (residues 15–23).

## Structural comparisons with other MG mAbs

Whether different MG mAbs bind nAChR through conserved or divergent mechanisms is an important question relevant to understanding the disease mechanism and developing therapeutics. To address this question, we compared the structure of Fab35 with that of two other MG mAbs (Fab198: PDB ID, 1FN4 and Fab192: PDB ID, 1C5D) (*Kontou et al., 2000*; *Poulas et al., 2001*). Interestingly, superposition of the structure of Fab198 onto that of Fab35 in the ternary complex shows that these two Fabs share not only a conserved immunoglobulin fold but also a similar antigen-binding site (*Figure 4a*). As such, the MIR loop fits well into the pocket surrounded by the CDR-H2, CDR-H3 and CDR-L3 loops of Fab198, as predicated by previous modeling studies (*Kleinjung et al., 2000*). The CDR-H2 loop of Fab198 is also in position to interact with the N-terminal helix adjacent to the MIR (*Figure 4b*). Even more remarkably, many key α1-binding residues in Fab35 are also conserved in Fab198 and they appear to make similar contacts to nAChR α1 in the modeled Fab198/nAChR α1 binding interface (*Figure 4a and c*). These residues include Trp47 (CDR-H2), Arg50 (CDR-H2), and Tyr95 (CDR-L3) at the center of the MIR-binding pocket, and Trp52 and Asp54 (both CDR-H2) which interact with the N-terminal helix. However, in contrast to the above structural similarities, the CDR-H3 loops between Fab198 and Fab35 differ significantly in length and sequence (*Figure 4b and c*). As a result, the CDR-H3 loop of Fab198 is too short to interact with the surface pocket of nAChR α1, which is, in the case of Fab35, occupied by the corresponding CDR-H3 loop (*Figure 4—figure supplement 1a*). These structural analyses suggest that mAb35 and mAb198 share a high similarity in binding mechanism to the core MIR/N-terminal helix region, but differ in the periphery of the binding interface.

On the other hand, superposition of the structure of Fab192 onto that of Fab35 in the ternary complex reveals substantial differences between them (*Figure 4—figure supplement 1b*). Although the constant domains ($C_H$ and $C_L$) of these two Fabs align very well structurally, the variable domains ($V_H$ and $V_L$) show a significant rotational twist, such that the MIR loop does not fit into the antigen-binding site of Fab192 (*Figure 4—figure supplement 1b*). Moreover, the key α1-binding residues of Fab35, such as Arg50 and Trp52 of CDR-H2, are not conserved in Fab192 (*Figure 4c*). This structural comparison suggests that Fab192 differs significantly from Fab35 in terms of their binding mechanisms to nAChR α1, confirming and extending the differences that were previously recognized between mAb35 and mAb192 (*Luo et al., 2009*).

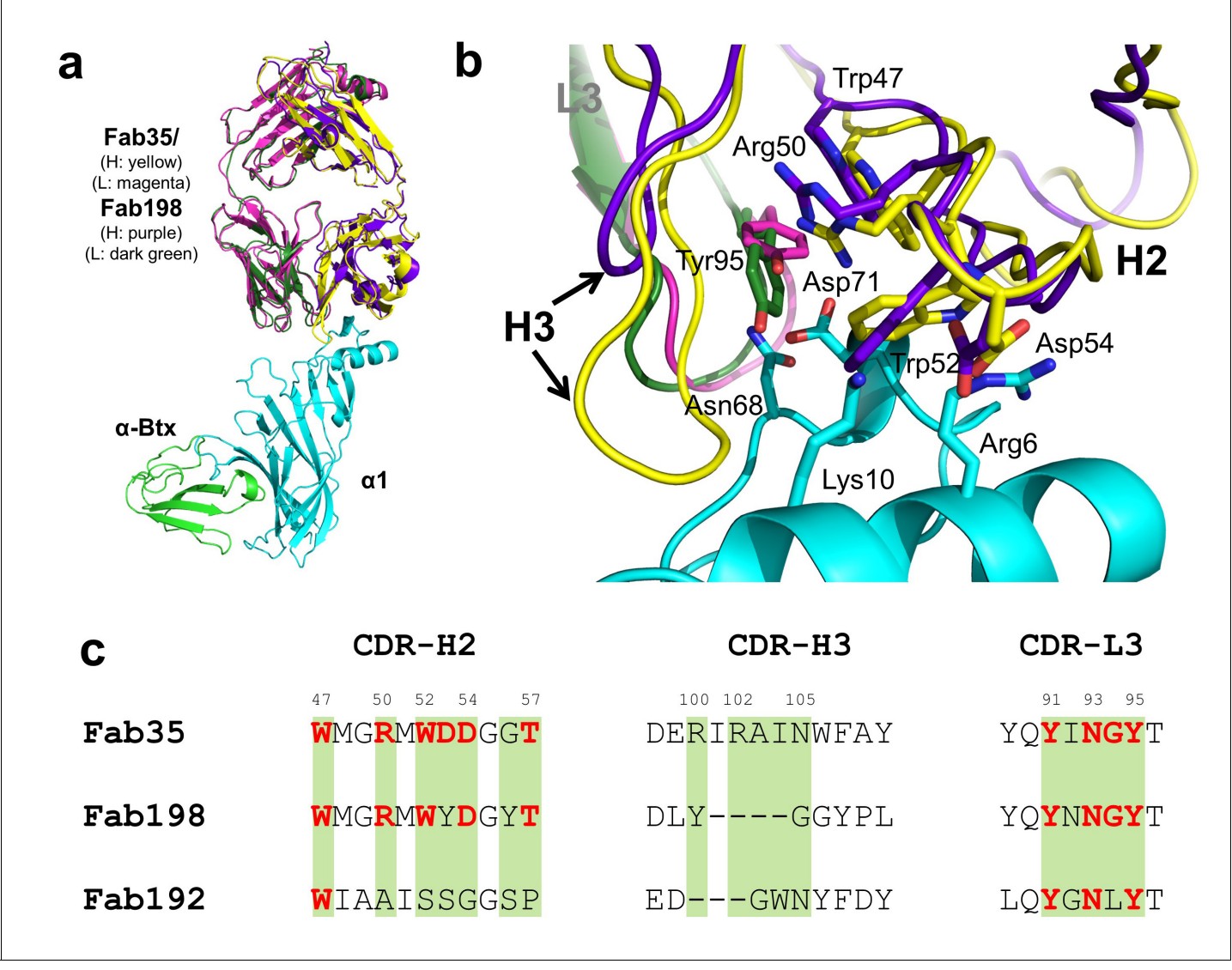

**Figure 4.** Structural comparisons among MG mAbs. (**a**) Superposition of Fab198 (*Poulas et al., 2001*) (heavy chain: purple and light chain: dark green) onto Fab35 in the Fab35/nAChR α1/α-Btx ternary complex using the $C_\alpha$ backbone. (**b**) Detailed comparison of the binding interface. The residues are colored according to their protein subunits. Note that key α1-binding residues in Fab35, including Trp47, Arg50, Trp52 and Asp54 of $V_H$ and Tyr95 of $V_L$ are conserved in Fab198, and seem to be able to make similar contacts to nAChR α1 in the modeled interface. The CDR-H3 loop of Fab198 (purple) is substantially shorter than that of Fab35 (yellow), as indicated by arrows. (**c**) Structure-based sequence alignment of the nAChR α1-binding loops (CDR-H2, CDR-H3 and CDR-L3) between Fab35, Fab198 and Fab192 (*Kontou et al., 2000*). Residues shaded in light green are involved in nAChR α1 binding in Fab35, some of these (in bold font and colored in red) are conserved in Fab198 or Fab192. Note that Fab35 and Fab198 share a high similarity in their nAChR α1-binding CDR-H2 and CDR-L3 loops, but differ significantly in CDR-H3. On the other hand, Fab192 differs significantly from Fab35 and Fab198, especially in the CDR-H2 and CDR-H3 loops (See also *Figure 4—figure supplement 1*).

The following figure supplement is available for figure 4:

**Figure supplement 1.** Structural comparison between Fab35 and Fab198/Fab192.

## Antibody-receptor binding specificity

Another important question of clinical and mechanistic relevance is the binding specificity of MG mAbs to different nAChR subunits and to nAChR α1 from different species. To address the first part of this question, Fab35-contacting residues were mapped onto aligned sequences of human nAChR subunits (*Figure 5a*). A subset of nAChR family members, including α2, α3, α5 and β3, has either

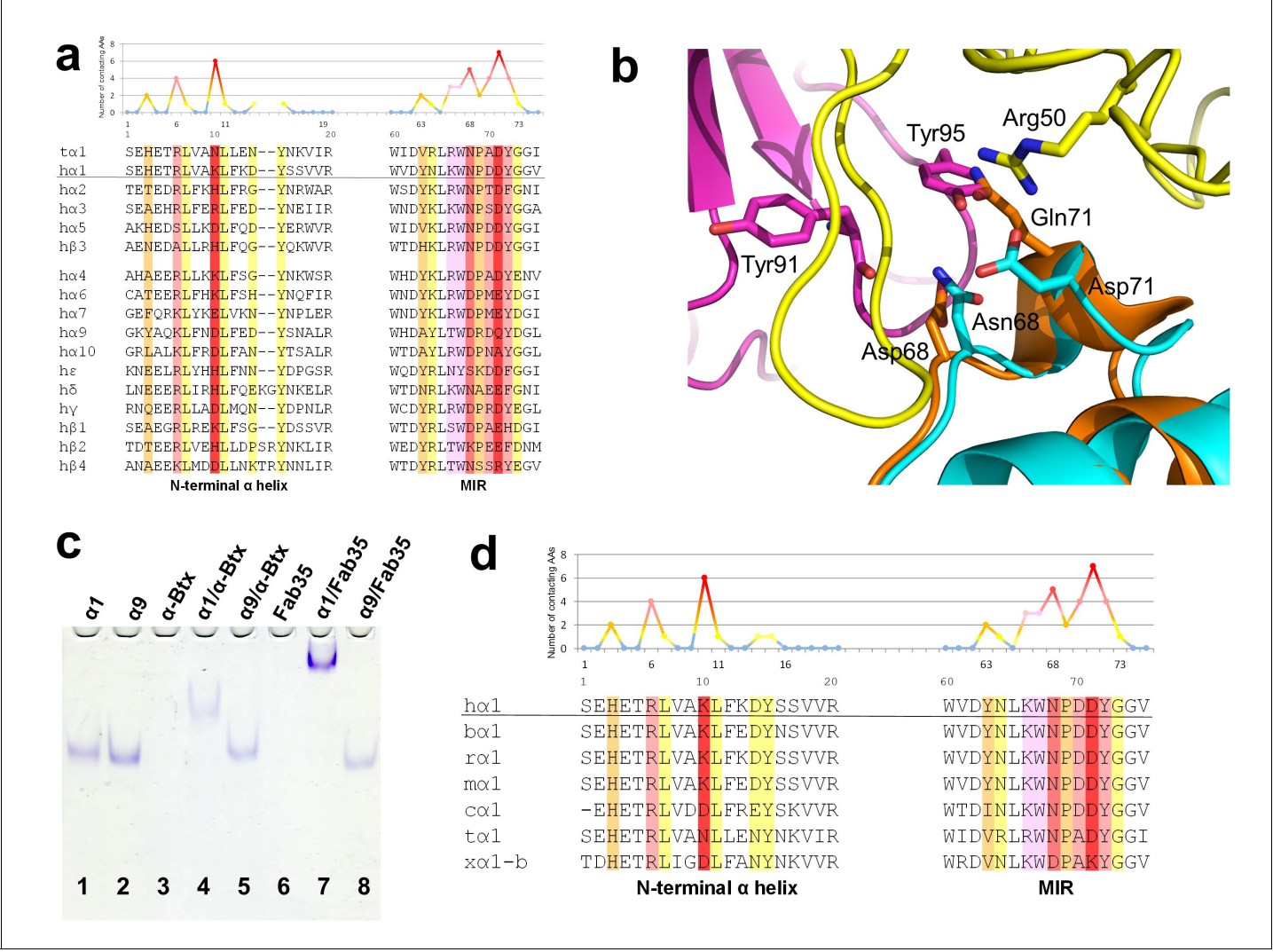

**Figure 5.** Specificity of antibody-receptor binding. (**a**) Multiple sequence alignment of the N-terminal α helix (left) and the MIR (right) of human nAChR family members. The sequence of human nAChR α1 (hα1) in the crystal structure is underlined. Abbreviation follows as t (torpedo) and h (human). The Fab35-contacting profile for human nAChR α1, indicating how many Fab35 residues are directly contacting with each particular residue of human nAChR α1, is shown above the sequence. The aligned sequences are colored based on the contacting profile, with red color indicating highly contacting residues ('hotspots'). (**b**) Superposition of nAChR α9 (orange) (*Zouridakis et al., 2014*) onto the nAChR α1 in the Fab35/nAChR α1/α-Btx ternary complex showing the disrupted binding interface. (**c**) Native PAGE showing the binding specificity of Fab35. Lane 1: α211 (labeled as α1), Lane 2: α9, Lane 3: α-Btx, Lane 4: α211 plus α-Btx, Lane 5: α9 plus α-Btx, Lane 6: Fab35, Lane 7: α211 plus Fab35, and Lane 8: α9 plus Fab35. Note that α-Btx in Lane 3 and Fab35 in Lane 6 were not visible because both proteins are positively charged and migrated upward under the native gel electrophoresis condition. Lanes with α-Btx were included as positive controls (Lanes 3–5). Lanes 4 and 5 show that both nAChR α1 and α9 bind α-Btx. Note that the α9/α-Btx complex has a smaller shift than the α1/α-Btx complex. Lanes 7 and 8 show that Fab35 binds α1 but not α9. (**d**) Multiple sequence alignment of the N-terminal α helix (left) and the MIR (right) of nAChR α1 from several species along with the Fab35-contacting profile as in (**a**). Abbreviation follows as b (bovine), r (rat), m (mouse), c (chicken), t (torpedo) and x (*Xenopus*).

identical or homologous residues at the key Fab35-binding positions. These analyses suggest that Fab35 and other MIR-directed mAbs may be able to bind these nAChR subunits. On the other hand, the other nAChR members have divergent sequences at the Fab35-binding sites. For example, the key Fab35-binding residues in the MIR of nAChR α1, Asn68 and Asp71, are replaced by Asp and Gln in nAChR α9, respectively. On the N-terminal helix of nAChR α1, the key Fab35-binding residues, Arg6 and Lys10, are also replaced by Lys and Asp in nAChR α9, respectively. Superposition of the recently solved structure of the nAChR α9 ECD (PDB ID, 4UY2) (*Zouridakis et al., 2014*) with

that of the α1 in the Fab35-bound complex showed that the sequence divergence could disrupt both the shape and chemical complementarity of the binding interface (*Figure 5b*). Consistent with the above structural analyses, we have shown by native PAGE that Fab35 indeed binds specifically to nAChR α1 but not α9 (*Figure 5c*).

For the second part of the question, a similar structure-based sequence comparison shows that Fab35-binding residues are conserved in nAChR α1 across a wide range of species (*Figure 5d*), which is consistent with the high cross-reactivity of autoantibodies in MG and EAMG between different species (*Tzartos et al., 1998*, *1982*; *Luo et al., 2009*; *Gomez et al., 1981*). However, in *Xenopus* nAChR α1-b, Asn68 and Asp71 are substituted by Asp and Lys, respectively. Based on our ternary structure, such N68D substitution would disrupt the hydrogen bonds between the side chain amide group of Asn68 and both the main chain carbonyl of Tyr91 and Gly94 in the CDR-L3 (*Figure 3b*). In addition, the D71K substitution would introduce positive charge repulsion with Arg50 in the CDR-H2 and numerous steric clashes (*Figure 3a*). Consistent with these structural observations, previous studies have shown that *Xenopus* nAChR α1 indeed does not bind Fab35 (*Saedi et al., 1990*).

## Models of higher-order antibody-receptor structure

The nAChR receptors are pentamers of identical or homologous subunits. To see how neighboring subunits to the α1 may affect the antibody-receptor interactions, we analyzed the binding of Fab35 to a nAChR pentamer by structure-based modeling using our previously solved structure of the α7/AChBP chimera (PDB ID, 3SQ9, 3.1 Å resolution) (*Li et al., 2011*) as the nAChR ECD pentamer (*Figure 6a and b*). This model suggests that Fab35, by binding to the extruding tip of an α1 subunit, makes no direct contact to the neighboring subunits. Moreover, Fab35 binds nAChR α1 at a site that is far away from the ligand-binding site, consistent with the observation that MIR-directed antibodies generally do not affect the channel function (*Tzartos et al., 1981*; *Gomez et al., 1981*; *Tamamizu et al., 1996*).

We also modeled the binding of a complete mAb35 (Fab+Fc) to the full-length nAChR pentamer using the structure of an intact IgG1 (PDB ID, 1IGY) (*Harris et al., 1998*), which is the same IgG1 subtype as mAb35, and the *Torpedo* AChR pentamer (PDB ID, 2BG9, 4 Å resolution) (*Unwin, 2005*) or the human α4β2 nicotinic receptor (PDB ID, 5KXI, 3.94 Å resolution) (*Morales-Perez et al., 2016*) as templates. The model built on the *Torpedo* AChR (*Figure 6—figure supplement 1a*) is very similar to that built on the α4β2 nAChR (*Figure 6c*). The primary differece between the models is that the N-terminal helix of the α1 subunit in the *Torpedo* AChR appears to adopt a different orientation from the conserved conformation adopted by the corresponding helix in a number of nAChR structures (*Unwin, 2005*; *Dellisanti et al., 2007a*; *Morales-Perez et al., 2016*). The source of this structural difference is currently unknown but our analysis shows that it has little effect on the overall structure of the modeled full antibody-receptor complex. The modeled complex structure shows that the antibody is projected away from the central pore of the receptor, consistent with previous EM analyses (*Beroukhim and Unwin, 1995*). After putting all of the molecular components on proper scale and orientation, our model of the full-length antibody/receptor complex suggests that, as a result of steric and geometric constraints, the two Fabs from a single mAb35 antibody are unable to bind to the two α1 subunits within the same nAChR pentamer (*Figure 6c*). Consequently, MG antibodies will bind two α1 subunits from different pentamers, thereby cross-linking the nAChRs. Our modeling analyses are consistent with previous sucrose density gradient studies showing that mAb35 and similar mAbs cannot bind the two α1 subunits within the same pentamer but can cross-link adjacent nAChR pentamers (*Conti-Tronconi et al., 1981*). Considering that each muscle nAChR pentamer contains two α1 subunits and the relatively high density of nAChRs at the postsynaptic membrane, such inter-pentamer cross-linking mediated by MG antibodies could lead to a super high-order of antibody/receptor complex at the neuromuscular junction. Previous studies have shown that the binding of nAChR by the divalent MG antibodies rather than the monovalent Fab fragment leads to accelerated degradation of the receptor proteins (*Drachman et al., 1978*). This observation suggests that cross-linking of the nAChRs is a critical step in receptor degradation. Our analyses suggest that receptor cross-linking is an intrinsic property of mAb35 and mAb35-like MG antibodies. This cross-linkig could lead to the formation of large antibody-receptor complexes that disrupt the structure and function of the neuromuscular junction and induce the degradation of the nicotinic receptor proteins.

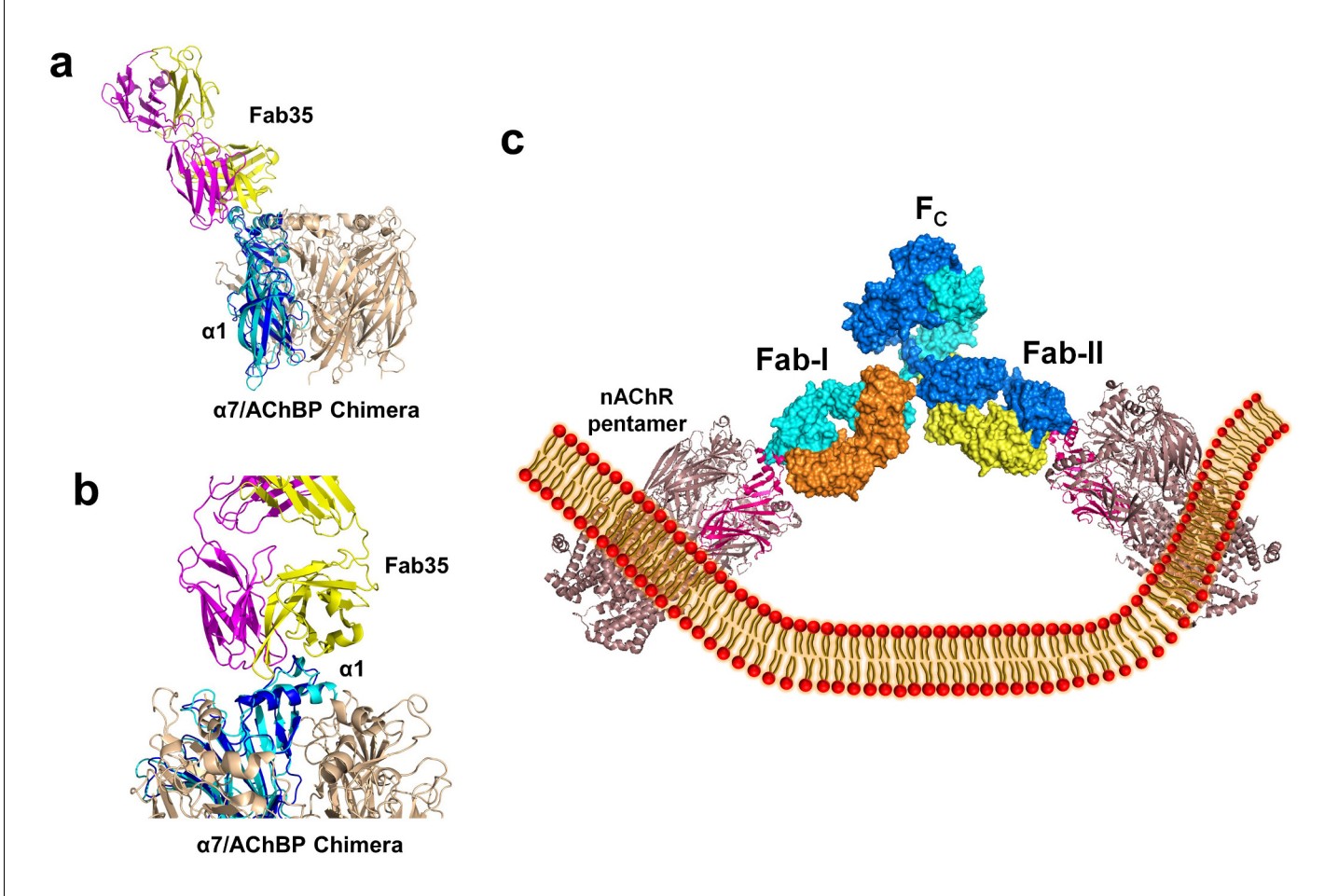

**Figure 6.** Modeling the binding of Fab35 to a nAChR pentamer. (**a**) Superposition of the Fab35/nAChR α1 ECD/α-Btx ternary complex on one subunit of the α7/AChBP chimera pentamer (blue) (PDB ID, 3SQ9) (*Li et al., 2011*) using the $C_\alpha$ backbone of ECDs as the reference. (**b**) Zoomed-in view of the contact between Fab35 and α7/AChBP Chimera. Fab35 makes no direct contact to the neighboring subunits in the pentamer. (**c**) Binding of a complete antibody (Fab+Fc) to a nAChR pentamer. The ternary Fab35/nAChR α1 ECD/α-Btx complex structure was used to guide the docking of an intact IgG1 antibody (PDB ID, 1IGY) (in surface model) (*Harris et al., 1998*) onto the human α4β2 nicotinic receptor (PDB ID, 5KXI, 3.94 Å resolution) (*Morales-Perez et al., 2016*). The two Fab domains (Fab-I and Fab-II) and the Fc region of IgG1 are indicated as shown. Each heavy chain is colored as blue and cyan. Each light chain is colored as yellow and orange. ECD of nAChR α1 is shown in magenta.

The following figure supplement is available for figure 6:

**Figure supplement 1.** Modeling the binding of a complete MG mAb to full-length nAChR(s).

## Discussion

Our crystal structure of Fab35 bound to the nAChR α1 ECD provides the first atomic view of the detailed interactions between an EAMG antibody and the nAChR. Our structure reveals that the MIR loop inserts deeply into the antigen-binding pocket of Fab35 and that the adjacent N-terminal helix makes extensive contacts with the CDR-H2 loop of the heavy chain. The binding interface structure and detailed interactions observed in the crystal can be cross-validated with existing biochemical data. Earlier studies mapped the core region of MIR to residues 67–76 (*Barkas et al., 1988*; *Tzartos et al., 1988*; *Das and Lindstrom, 1989*). More recent studies using natively folded nAChR α1/α7 chimera proteins (*Luo et al., 2009*) or GFP-fused protein fragments (*Morell et al., 2014*) showed that the N-terminal helix (residues 1–14) is also important for high-affinity MG antibody binding. These studies further indicated that other regions of nAChR, including the loop following helix 1 (residues 15–32) (*Luo et al., 2009*) and the β5-β6-loop packing against the MIR (residues

110–115) (*Morell et al., 2014*), also contribute to the binding of some MG antibodies. Our structures reveal that the antibody-receptor binding interface indeed centers on the MIR and the N-terminal helix and also includes peripheral regions such as residues 15–23. Although the *β5-β6*-loop (residues 110–115) does not contact Fab35 in our structure, it may contribute to antibody binding indirectly by maintaining the natively folded structure of MIR (*Morell et al., 2014*).

Specific residues on MIR (*Papadouli et al., 1990*, *1993*; *Bellone et al., 1989*) and the N-terminal helix (*Morell et al., 2014*) have also been analyzed using peptide/protein fragments containing mutations at specific positions for their roles on MG antibody binding. These studies showed that Asn68 and Asp71 of MIR are essential for MG antibody binding while the surrounding residues including Pro69 and Tyr72 showed partial effect when mutated. The essential role of Asn68 and Asp71 was further confirmed by site-directed mutagenesis of N68D and D71K in the intact receptor (*Saedi et al., 1990*). On the N-terminal helix of *Torpedo* nAChR α1, two exposed residues, Arg6 and Asn10, which correspond to Arg6 and Lys10 in human nAChR α1, respectively, are found to be critical to MG antibody binding by mutational analyses (*Morell et al., 2014*). Most of the nAChR residues found to be essential for antibody binding by mutagenesis studies, including Asn68 and Asp71 from the MIR and Arg6 and Lys10 from the N-terminal helix, indeed correspond to interaction 'hotspots' at the Fab35/nAChR α1 interface. These 'hotspot' residues anchor multiple interaction networks at the interface and make extensive contacts to the antibody. Although biochemical mapping of antibody-binding residues on nAChR α1 were performed with different antibodies (e.g. mAb210 and mAb132A) (*Barkas et al., 1988*; *Tzartos et al., 1988*; *Das and Lindstrom, 1989*; *Saedi et al., 1990*; *Papadouli et al., 1990*, *1993*; *Luo et al., 2009*; *Morell et al., 2014*), it is remarkable that these biochemical data agree so well with our crystal structure, suggesting that many MIR-directed antibodies may share high similarities in their binding sites on the nAChR. It has been suggested that EAMG and MG antibodies may bind epitopes different from MIR and that these MG antibodies may be competed off by mAb35 through steric effect rather than direct epitope competition (*Luo et al., 2009*). Our structures (*Figure 1* and *Figure 6c*) indeed support this possibility, which is represented by mAb192 (also see below). However, our structural analyses also reveal that many MIR residues at the center of the antibody-receptor are important for the high affinity binding of a variety of MG antibodies (e.g., mAb35, mAb210, and mAb132A). This is a rather surprising finding given the potential heterogeneity of nAChR antibodies mentioned above. An important implication of this finding is that molecular mimicries of the MIR and its immediate surrounding regions could be developed to bind a significant fraction of MIR-directed MG autoantibodies. Such molecules could be useful leads for developing diagnostics and therapeutics for MG (*Tzartos et al., 1998*; *Masuda et al., 2012*; *Sophianos and Tzartos, 1989*).

MIR-directed antibodies display a wide range of binding properties. Some such antibodies (e.g. mAb35) exclusively bind natively folded receptor and others (e.g. mAb210) are also capable of binding denatured receptor or isolated MIR peptides (*Luo et al., 2009*; *Morell et al., 2014*). The binding mechanisms between different MG antibodies may have subtle or significant differences. Our comparative structural analyses indicate that Fab35 and Fab198 share highly similar binding mechanisms to nAChR α1, especially in the MIR/N-terminal helix core region. On the other hand, Fab192 seems to have very different nAChR-binding mechanisms from Fab35 and Fab198 even though Fab192 can be competed off by mAb35 through steric effect. Most MIR-directed mAbs, such as mAb35, bind preferably to folded receptors. Our structure shows that the complete epitope consists of two separated peptide regions (the MIR loop and the N-terminal helix) that are required to fold together properly for optimal binding. These structural observations are consistent with previous studies of chimeras and mutants showing that the native conformation of the MIR that permits binding of mAbs 35 and 198 depends on the interaction of the N-terminal helix with the MIR loop (*Luo et al., 2009*; *Dellisanti et al., 2007a*). Moreover, for mAb35, a significant amount of binding energy may derive from the insertion of its CDR-H3 loop to the surface pocket on nAChR α1, whose structure can only form in the natively folded receptor. This unique feature of mAb35 is consistent with the observation that mAb35 is particularly conformation sensitive in binding to nAChR α1 (*Luo et al., 2009*).

Our modeling analyses indicate that neighboring subunits in the nAChR pentamer do not make direct contact to Fab35, suggesting that the binding interactions observed in our crystal structures represent most, if not all, of those in the native complexes between mAb35 and the full-length nAChR pentamer. However, it is known that antibodies in MG serum and mAbs bind mature and

pentameric nAChR more tightly than the unassembled nAChR α1 monomer (*Luo et al., 2009*; *Merlie and Lindstrom, 1983*; *Conroy et al., 1990*), and there is evidence that MIR plays an important role in initiating conformational maturation of subunits prior to their assembly into pentameric AChR receptors (*Luo et al., 2009*). Structural differences between monomeric α1 and α1 in a muscle nAChR heteropentamer could account for the different binding affinities. Although the N-terminal helix of the α1 subunit in the full length *Torpedo* nAChR has a significantly different orientation from that in the monomeric mouse/human α1 crystal structures (*Unwin, 2005*; *Dellisanti et al., 2007a*), it is not clear if this structural difference, which may be due to the limited resolution (4 Å) of the full length *Torpedo* nAChR, is realistic and hence responsible for the different binding affinities. However, we cannot rule out the possibility that subtle structural differences between the monomeric and pentameric forms of mammalian nAChR α1 could affect the binding affinity, especially for MG autoantibodies, like mAb35, that are particularly conformation sensitive. Furthermore, the monomeric α1 may be more dynamic than its counterpart in the fully assembled pentamer, and the conformational flexibility could reduce the binding affinity through entropic effects.

Our studies reveal two structural insights into the antibody-nAChR interaction that may have important mechanistic and clinical implications. The first is the conservation of the key Fab35-binding residues in a number of nAChR family members, including α2, α3, α5 and β3. This observation suggests potential cross-reactivity of α1-derived MG antibodies to these nAChR family members. As discussed above, because many MIR-directed mAbs share the same binding residues on nAChR α1, this cross-reactivity may not be limited to mAb35, but can also occur among other MG autoantibodies. Consistent with this notion, it has been shown that some MG mAbs bind neuronal nAChR subunits. mAb35 was shown to bind chicken nAChR α3 and also suggested to bind human α2, α3, α5 and β3 (*Conroy and Berg, 1995*). Another MIR-directed antibody, mAb210, has been used to bind human α5 and β3 (*Kuryatov et al., 2008*; *Wang et al., 1996*). The wide expression and diverse physiological functions of nAChR members within and outside the neuronal system are being increasingly recognized, raising an intriguing question whether the cross-reactivity of nAChR with autoantibodies has broader pathological effects than currently recognized.

The second is the structural basis of antibody-mediated receptor cross-linking. Our crystal structures reveal a well-defined orientation of the bound antibody with respect to the receptor due to the relatively rigid binding interface. Based on this structural feature, we modeled the complex of full-length nAChR pentamer bound by the intact MG antibody, which suggests that MG antibodies are unlikely to bind the two α1 subunits within the same muscle nAChR pentamer, but rather two α1 subunits from different pentamers, thereby cross-linking the nAChR receptors (*Tzartos et al., 1981*; *Conti-Tronconi et al., 1981*). These modeling analyses provide a structural support for previous functional observations that MG antibody-mediated nAChR cross-linking accelerates the degradation of the receptor proteins (*Drachman et al., 1978*). We further noticed that MG antibodies are unlikely to bind two nAChR pentamers oriented vertically on a flat membrane surface even when considering the hinge flexibility of the antibody (*Figure 6—figure supplement 1b*). Assuming that the nAChR pentamers in the membrane cannot be tilted freely, a potential effect of antibody-mediated receptor cross-linking is membrane curvature. This raises an intriguing question if such distortion of the membrane structure could play a role in the internalization and degradation of nAChRs at the neuromuscular junction (*Figure 6—figure supplement 1c*). While the detailed mechanisms by which antibody-mediated receptor cross-linking induces the receptor degradation remain to be elucidated, molecular mimicries of the MIR should prevent such cross-linking and degradation of nAChR by competitive binding to MG antibodies (*Sophianos and Tzartos, 1989*). Our studies suggest that it is possible to develop drug molecules to inhibit the binding of a large fraction of MG antibodies to nAChR and related pathological immune reactions, and the crystal structures presented here provide a basis for developing such drug molecules.

## Materials and methods

### Construction of the stabilized nAChR α1 ECD (= α211)

The mouse α211 construct was provided by Dr. Zuo-Zhong Wang, Zilkha Neurogenetic Institute, Department of Cell and Neurobiology, Keck School of Medicine, University of Southern California. The detailed construction information is as previously described (*Yao et al., 2002*). Briefly, a Flag-

tag and a His-tag were added at the N-terminus and the C-terminus, respectively, for higher expression and purification purposes. The construct was truncated at the 211th amino acid from the N-terminal of nAChR α1 subunit without a signal sequence, and three point mutations (V8E, W149R, and V155A) were introduced for improved solubility and stability (*Dellisanti et al., 2007a*; *Chen, 2010*). The human α211 construct was designed based on the mouse α211 construct, and the synthesized cDNA was ordered from GenScript. The gene codons were optimized for yeast, and cloned into a pPICZαA vector using the EcoRI and XbaI sites.

## α211 expression

Both mouse and human α211 constructs were linearized by digesting with SacI restriction enzyme (New England Biolabs) and transformed into KM71H of *P. pastoris* (Invitrogen) by electroporation. The transformants were plated on YPDS plates, which contained 100 μg mL$^{-1}$ Zeocin. Plates were incubated at 30°C for 3–5 days until colonies formed, and several colonies were restreaked on fresh YPDS plates. Pre-inoculation was made by seeding a single colony in 30 mL BMGY medium. The culture was incubated at 30°C with shaking overnight. About 5–7 mL of this culture was used to inoculate 500 mL of BMGY in a 2 L baffled flask (total 2 L of culture). The inoculated culture was incubated at 30°C with shaking to OD$_{600}$ value of 6. Cells were harvested by centrifugation at 3000×g for 15 min at room temperature. The supernatant was discarded, and cell pellets were resuspended in 400 mL of BMMY medium for induction. The resuspended culture was divided between two 2 L baffled flasks (200 mL each) and incubated at 20°C with shaking for 72 hr. 100% methanol was added every 24 hr to a final concentration of 0.5% (v/v) to induce protein expression. After 72 hr of induction, cells were harvested by centrifuging at 6000×g for 20 min at room temperature. Protein purification proceeded with the supernatant as the protein was secreted.

## α211 purification

Ni-NTA agarose beads (QIAGEN) were incubated with the supernatant at 4°C overnight with end-over-end rotation. The protein was eluted with elution buffer (50 mM NaH$_2$PO$_4$, pH 7.8, 0.5 M KCl, 10% (v/v) glycerol, and 500 mM imidazole) after washing with washing buffer containing 20 mM imidazole and 0.1% Triton X-100 to remove loosely bound proteins. The eluted protein was concentrated and ran over a size exclusion column (Superdex 75 10/300 GL, GE Healthcare) with 20 mM HEPES, pH 7.5 and 150 mM NaCl buffer for further purification. After each peak fraction was analyzed by OD$_{280}$ measurement and SDS-PAGE, fractions containing α211 were pooled and concentrated for further experiments.

## Cell culture and reagents for mAb35

Hybridoma cells of mAb35 were purchased from American Type Culture Collection (ATCC). The cells were maintained in DMEM medium containing 1.97 g L$^{-1}$ NaHCO$_3$ and 10% fetal bovine serum (FBS). The cells were cultured in a 37°C incubator with 5% CO$_2$ and subcultured every 2 to 3 days with cell density between $1 \times 10^5$ and $1 \times 10^6$ cells mL$^{-1}$. For protein production, the cell culture was incubated at 37°C for several days until the medium color changed to yellow.

## mAb35 purification

After 7–10 days of incubation at 37°C, the cell culture was harvested by centrifuging at 6000×g for 15 min. Affinity purification was performed using Protein G Sepharose 4 Fast Flow (GE Healthcare). The supernatant and beads were incubated at room temperature for 2 hr with rotation. The beads were washed with washing buffer (20 mM sodium phosphate, pH 7.0), and the protein was eluted with elution buffer (0.1 M glycine-HCl, pH 2.7). Due to the low pH of the elution buffer, a neutralizing buffer (1 M Tris-HCl, pH 9.0) was added to the collection tubes (60 to 200 μL mL$^{-1}$ elute) prior to collection. After checking the presence of the protein with SDS-PAGE gels, the protein elution was concentrated and ran over a size exclusion column (Superdex 200 10/300 GL, GE Healthcare). The fractions of protein peak were pooled and concentrated for further study.

## mAb35 digestion and Fab35 purification

Purified mAb35 was buffer-exchanged into digestion buffer (20 mM sodium phosphate, pH 7.0, 10 mM EDTA and 20 mM cysteine-HCl; adjust pH to 7.0 right before use) using Zeba Spin columns

(Thermo Scientific), and the resulting sample was incubated with immobilized papain beads (Thermo Scientific). The sample was rotated at 30°C overnight, and the flow through was collected. The protein was identified by SDS-PAGE, and then the sample was concentrated down to ~500 µL while exchanging buffer to Mono Q Buffer A (20 mM sodium phosphate, pH 7.0). An anion exchange column (Mono Q HR 5/5, GE Healthcare) was used to separate the Fab portion from the rest using a salt gradient of 0–100% Buffer B (20 mM sodium phosphate, pH 7.0 and 1 M NaCl). Fab fractions (Fab35) were collected and concentrated for gel shift assay and α211 complex purification.

## Gel shift assay

α211, α-bungarotoxin (α-Btx) and Fab35 were mixed in an equimolar ratio, and the mixture was incubated on ice for 1 hr. For the binding specificity experiment, α9-3Mut, which contains three point mutations (V189W, S191F and G193S) in the loop C that were designed to enhance the binding of α-Btx, was prepared in the same manner as α211 (Kaori Noridomi, Ph.D. dissertation, University of Southern California, 2015). A 10% native PAGE gel was run at 4°C for 3.5 hr at 100–120 V, ~15 mA, with 1× TBE buffer. Bands were detected with Coomassie Blue staining.

## The α211/Fab35/α-bungarotoxin complex purification

α211, α-Btx and Fab35 were mixed at a 1:1.5:1.5 molar ratio, and the mixture was incubated on ice for 1 hr. Ni-NTA purification was performed to remove excess α-Btx and Fab35. The elution was run over a size exclusion column (Superdex 200 10/300 GL, GE Healthcare) with 20 mM HEPES, pH 7.5 and 150 mM NaCl buffer. Two peaks were obtained, and fractions of each peak were pooled separately and concentrated for crystallization.

## Crystallization and X-ray diffraction data collection

The purified and concentrated ternary complex of α211 (both mouse and human)/Fab35/α-Btx was diluted to 2.5 mg mL$^{-1}$. Crystals were screened using Crystal Screen, Crystal Screen 2 and Index kits (Hampton Research) by a hanging drop method at room temperature. Each reservoir contained 0.5 mL of screening solution, and each drop contained 0.5 µL of protein and 0.5 µL of reservoir solution. Three conditions gave initial hits, and one condition was selected to optimize a crystallizing condition. The optimized condition was 0.1 M sodium cacodylate trihydrate, pH 6.5, 0.1–0.15 M calcium acetate hydrate and 18–20% (w/v) PEG 8K. Rod-shaped crystals were grown as bundle with the size of 10 µm × 20–100 µm × 200 µm. Crystals were harvested using harvest solution (0.2 M calcium acetate hydrate, 0.1 M sodium cacodylate trihydrate, pH 6.5 and 30% (w/v) PEG 8K) and cryo-solution (harvest solution +20% glycerol) as follows. Crystals were transferred to different concentrations of harvest/cryo mixture solution to protect crystals from osmotic shock (in order of 100% harvest solution, 3:1 harvest/cryo solution, 1:1 harvest/cryo solution, 1:3 harvest/cryo solution and 100% cryo solution). Each incubation time was approximately 5–10 min. The cryoprotected crystals were fished using 100-300 µm Hampton CryoLoop (Hampton Research) and flash-cooled in liquid nitrogen.

Diffraction data were collected at the Advanced Photon Source (APS) beamline 23-ID-B at Argonne National Laboratory using a 10 µm × 10 µm beam (λ = 1.0332 Å, 12.000 keV) with the attenuation factor of 5.0 and a MARmosaic 300 CCD detector. The detector distance was 300.0 mm. The oscillation range and the exposure time per frame were 0.5° and 2.0 s, respectively. Data were processed and scaled using HKL2000 package (*Otwinowski and Minor, 1997*). Both human and mouse complex crystals belong to the space group C2, with unit cell dimensions of a = 160.0 Å, b = 42.1 Å, c = 136.5 Å, β = 117.1°; and a = 159.9 Å, b = 42.0 Å, c = 137.6 Å, β = 116.5°, respectively. The crystal structures were solved by molecular replacement using PHASER MR (*McCoy et al., 2007*) in CCP4 (*Collaborative Computational Project Number 4, 1994*) and the coordinates of α211/α-Btx complex (PDB ID, 2QC1) and Fab198 (PDB ID, 1FN4, modified to poly-Ala) (*Dellisanti et al., 2007a*; *Poulas et al., 2001*). Refmac5 was used for the final refinement of the structures (*Murshudov et al., 2011*, *1997*). The amino acid sequence of mAb35 was obtained from an antibody sequencing company, MCLAB Molecular Laboratories (San Francisco, CA), and the sequence of Fab35 was added using Coot (*Emsley et al., 2010*). Additional model building in Fab35 was carried out with O (*Jones et al., 1991*). Crystallographic analysis and refinement statistics are summarized in *Supplementary file 1*.

## Accession numbers

Coordinates and structure factors have been deposited in the Protein Data Bank under accession codes PDB: 5HBT (Fab35/human nAChR α1 ECD/α-Btx) and 5HBV (Fab35/mouse nAChR α1 ECD/α-Btx).

## Acknowledgements

This research used resources of the Advanced Photon Source, a US Department of Energy (DOE) Office of Science User Facility operated for the DOE Office of Science by Argonne National Laboratory under Contract No. DE-AC02-06CH11357. We would like to thank the GM/CA staff members (beamline 23-ID), especially Ruslan Sanishvili for help with data collection. We thank the USC Nano-Biophysics Core Facility and Shuxing Li for providing crystallography equipment. We thank Richard Roberts and members of his lab, Aaron Niclols, Lan Huong Lai, and John Mac, for proofreading the manuscript. We also appreciate Reza Kalhor for his discussion and comments on the manuscript. This work was supported in part by US National Institutes of Health grants, R01 GM064642 (LC) and R01 AI113009 (LC).

## Additional information

### Funding

| Funder | Author |
| --- | --- |
| National Institutes of Health | Lin Chen |

The funders had no role in study design, data collection and interpretation, or the decision to submit the work for publication.

### Author contributions

KN, Data curation, Writing—original draft, Writing—review and editing, Wrote and revised the paper, Designed all experiments, Carried out all aspects of experiments and collected data, Solved and analyzed structures; GW, Harvested crystals and collected data, Assisted with solving structures, Assisted with preparing and editing the manuscript; MNH, Assisted with optimizing a receptor expression. Assisted with purifying receptors and mAbs/Fabs; GWH, Assisted with refining and verifying structures; LC, Wrote and revised the paper, Designed all experiments, Solved and analyzed structures, Supervised the project

### Author ORCIDs

Kaori Noridomi, http://orcid.org/0000-0002-4303-7511
Go Watanabe, http://orcid.org/0000-0003-3911-3021
Lin Chen, http://orcid.org/0000-0003-4798-6199

## Additional files

### Supplementary files

• Supplementary file 1. The statistics of data collection and structure refinement for Fab35/ human nAChR α1 ECD/α-Btx ternary complexes and Fab35/mouse nAChR α1 ECD/α-Btx ternary complexes.

• Supplementary file 2. Contacting residues at the Fab35/human nAChR α1 ECD interface. Residues at the Fab35/human nAChR α1 (α211) interface were mapped using the contact program in CCP4 with a 4.5 Å distance cutoff (*Winn et al., 2011*). For each residue of nAChR α1 involved in antibody binding (Chain B/α211 listed in the first column), its interacting residues from the light chain (Chain C listed in the second column) and the heavy chain (Chain D listed in the third column) of Fab35 are listed in the corresponding row. Residues on the N-terminal helix is highlighted in yellow and residues on the MIR loop is highlighted in light green. Note that several nAChR α1 residues, including R6, K10, N68 and D71/Y72 (red font), contact a large number of antibody residues. These residues

can be considered as 'hotspots' of the binding interface. Y72 can be considered as part of the Asp71 'hotspot' (see the text).

## Major datasets

The following datasets were generated:

| Author(s) | Year | Dataset title | Dataset URL | Database, license, and accessibility information |
|-----------|------|---------------|-------------|--------------------------------------------------|
| Noridomi K, Watanabe G, Hansen MN, Han GW, Chen L | 2017 | Complex structure of Fab35 and mouse nAChR alpha1 | http://www.rcsb.org/pdb/explore/explore.do?structureId=5HBV | Publicly available at the RCSB Protein Data Bank (accession no: 5HBV) |
| Noridomi K, Watanabe G, Hansen MN, Han GW, Chen L | 2017 | Complex structure of Fab35 and human nAChR alpha1 | http://www.rcsb.org/pdb/explore/explore.do?structureId=5HBT | Publicly available at the RCSB Protein Data Bank (accession no: 5HBT) |

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
