## [Decision Letter]

Thank you for submitting your article "Structural insights into the molecular mechanisms of Myasthenia Gravis and therapeutic implications" for consideration by *eLife*. Your article has been reviewed by two peer reviewers, and the evaluation has been overseen by Kenton Swartz as the Reviewing Editor and Richard Aldrich as the Senior Editor. The following individual involved in review of your submission has agreed to reveal his identity: Jon M Lindstrom (Reviewer #2).

The reviewers have discussed the reviews with one another and the Reviewing Editor has drafted this decision to help you prepare a revised submission.

Summary:

This manuscript from Lin Chen's group describes structural efforts to understand the interactions of an antibody with the nicotinic receptor that can trigger myasthenia gravis (MG). The manuscript presents the first high resolution structure of a soluble extracellular domain of the alpha1 subunit of the nicotinic receptor in complex with a Fab fragment from the standard antibody known to cause experimental myasthenia gravis. The authors use the structure of this complex as a basis for speculation on the relative significance of different atomic interactions at the receptor-Fab interface in determining subunit specificity, and perform sequence comparisons and qualitative binding assays by native PAGE to test binding to the soluble extracellular domain of the alpha1 and alpha9 subunits. The authors then present a speculative mechanism suggesting that cross-linking of pentamers by the whole IgG could trigger membrane deformation, resulting in receptor internalization and eventually myasthenic syndrome.

The strengths of this manuscript and underlying work are in the introductory and structural sections. The authors lay out nicely the physiological background and significance, setting up the structural findings effectively. This piece of the study, alone, is impactful. We have some concerns about the refinement details, described below, but we do not expect those issues to change the overall interpretations or conclusions made in the manuscript. The following are essential revisions that are needed to improve the presentation.

Essential revisions:

1) The crystallographic statistics raise two issues that need to be addressed. First, for both datasets, there is an unacceptably large spread between R and Rfree (8.5-10%). This large spread suggests a high degree of model bias and inappropriate restraints used during refinement. This large spread is surprising given the relatively small spread between R and Rfree in the original alpha1-bungarotoxin complex structure from the same group (2.4% difference, PMID 17643119). Second, the Ramachandran statistics are poor; >90% of the residues should be in the favored region, and good justification should be made for any residues in the outlier region. Strong electron density and/or comparison with higher resolution structures is a requirement for confidence in modeling a residue in an outlier conformation. Thus, regions with poor electron density should have better Ramachandran statistics, not worse, as suggested by footnote b in [Supplementary-material SD1-data]. The authors need to address these two issues either by improving the quality of the refined model or explaining why the statistics, as they currently stand, are justified. One way to improve model quality without too much effort might be using a reference model during refinement (phenix.refine makes this reasonably straightforward).

2) The mechanism suggested for pentamer cross-linking inducing curvature and thereby internalization and pathology is very speculative and needs to be toned down. The reference cited in the Discussion (Drachmann et al., 1978) on cross-linking and curvature does not, as far as we see, actually propose, test or validate a curvature mechanism. As the authors discuss, modeling of cross-linked pentamers is tentative, in part because it relies on the older EM model of the Torpedo receptor, which has a 'funny' alpha1 helix conformation compared to all other structures, and in part because Fabs/IgG's have a great deal of inherent flexibility. We request that you move Figure 6 and 7 to a single supplemental figure and to limit presentation of this hypothetical mechanism to a single paragraph in the Discussion section.

3) The manuscript would have been strengthened by the addition of mutants and binding assays to specifically interrogate the interface seen in the X-ray structure. Although we appreciate that this would require considerable additional work, the authors can and should bring in the literature to help support the interface they see in the structure, as Luo et al., 2009 and others have predicted some of the interactions seen in the present study.

[Editors' note: further revisions were requested prior to acceptance, as described below.]

Thank you for resubmitting your work entitled "Structural insights into the molecular mechanisms of Myasthenia Gravis and therapeutic implications" for further consideration at *eLife*. Your revised article has been favorably evaluated by Kenton Swartz as Reviewing Editor and Richard Aldrich as Senior editor, and two reviewers.

The manuscript has been improved but there are some remaining issues that need to be addressed before acceptance, as outlined below:

1) There is something funny going in paragraph four of the Discussion section. A part of one sentence appears to have been deleted.

2) It would be helpful for the authors to include a few remarks on the following two issues:

Although mAb 35 and mAb 198 are very similar in structure, other EAMG and MG antibodies that bind to this region may not be so similar or bind to identical epitopes. The high frequency of MIR antibodies in MG and EAMG sera was determined by competitive binding between mAbs with serum antibodies. Figure 1 clearly illustrates how large Fab 35 is with respect to the extracellular domain of α 1. Large intact mAb 35 is likely to compete with intact large serum autoantibodies for binding in this region to nearby but somewhat different epitopes than are recognized by mAb 35. mAB 192 is a similar case. The relative size of an antibody and an AChR is nicely illustrated in Figure 6. The variety of epitopes in the main immunogenic region is an important factor relevant to the Discussion, paragraph two, about trying to treat MG with mimotopes or for developing epitope-specific diagnostics.

In the Results section it is noted that the N terminal α helix of α 1 in Torpedo AChR differs from that in Bellone, Tang, Milius and Conti-Tronconi, 1989. It is worth noting that Morell et al., 2014 shows that variations in MIR structure have large effects on ACh sensitivity. AChR sensitivity involves both ACh binding sites and transmembrane regions. The transmembrane regions are greatly effected by lipid. Solubilization of Torpedo AChRs puts them in an inactive conformation that can be corrected by reconstitution into the proper lipid composition (C daCosta and J Baenziger (2013) Cell Structure 21: 1271-1283). Most crystallized receptors are solubilized and consequently contact lipids partially or completely removed. The 5-HT3 receptor after solubilization was crystallized bound to nano bodies directed at its C loops that function as potent antagonists (Hasssaine et al. (2014) Nature 512: 276-281). This too may have global conformation consequences. Torpedo AChRs are unique in having had their structures determined in their native membrane. Of course, artifacts in their structural determination might also occur.

---

## [Author Response]

*Essential revisions:*

*1) The crystallographic statistics raise two issues that need to be addressed. First, for both datasets, there is an unacceptably large spread between R and Rfree (8.5-10%). This large spread suggests a high degree of model bias and inappropriate restraints used during refinement. This large spread is surprising given the relatively small spread between R and Rfree in the original alpha1-bungarotoxin complex structure from the same group (2.4% difference, PMID 17643119). Second, the Ramachandran statistics are poor; >90% of the residues should be in the favored region, and good justification should be made for any residues in the outlier region. Strong electron density and/or comparison with higher resolution structures is a requirement for confidence in modeling a residue in an outlier conformation. Thus, regions with poor electron density should have better Ramachandran statistics, not worse, as suggested by footnote b in [Supplementary-material SD1-data]. The authors need to address these two issues either by improving the quality of the refined model or explaining why the statistics, as they currently stand, are justified. One way to improve model quality without too much effort might be using a reference model during refinement (phenix.refine makes this reasonably straightforward).*

We thank the referee for the valuable comments. We reprocessed human and mouse nAChR α1 ECD/α-Btx data up to 2.60 and 2.70 Å, respectively. The new data stats are listed in the Table 1. This time, we include the CC_1/2_ (1)of the highest resolution shell. They are 0.568 and 0.537 for human and mouse nAChR α1 ECD/α-Btx data, respectively. We used multiple refinement programs (Phenix, autoBuster, and Remac5) to refine structures, and the final TLS and restrained refinement was performed with Refmac5 up to 2.61 Å resolution (*R*_work_/*R*_free_=20.7/26.0) for Fab35/Human nAChR α1 ECD/α-Btx and 2.70 Å resolution (*R*_work_/*R*_free_=22.8/26.9) for Fab35/Mouse nAChR α1 ECD/α-Btx. The R and R_free_ spread is reduced to 4.1-5.3% [Previous R and R_free_ gap was significantly large (8.5-10%).] The refinement stats are improved as well. Ramachandran statistics defined in MolProbity (2)shows 95.8 and 95.6% (> 90%) of residues are in the favored region. There is no Ramachandran outliers.

*2) The mechanism suggested for pentamer cross-linking inducing curvature and thereby internalization and pathology is very speculative and needs to be toned down. The reference cited in the Discussion (Drachmann et al., 1978) on cross-linking and curvature does not, as far as we see, actually propose, test or validate a curvature mechanism. As the authors discuss, modeling of cross-linked pentamers is tentative, in part because it relies on the older EM model of the Torpedo receptor, which has a 'funny' alpha1 helix conformation compared to all other structures, and in part because Fabs/IgG's have a great deal of inherent flexibility. We request that you move Figure 6 and 7 to a single supplemental figure and to limit presentation of this hypothetical mechanism to a single paragraph in the Discussion.*

We appreciate the reviewer’s comments on the speculative nature of our hypothesis that nAChR pentamer cross-linking induces membrane curvature and receptor internalization. In our modeling analyses, we have considered the limited quality of the EM model of the Torpedo receptor and the unusual conformation of the alpha1 helix, and also the flexibility of the antibody structure. We are reasonably confident about the main structural features of the constructed model as discussed in the manuscript. However, in the absence of any experimental data and precedents, we agree that the hypothesis is indeed speculative. Reference #4 is cited to show previous studies indicating that the binding of nAChR by divalent MG antibodies rather than the monovalent Fab fragment leads to accelerated degradation of the receptor proteins. This is meant to support the idea that cross-linking of the nAChRs is a critical step in receptor degradation, not to test and validate a curvature mechanism. We have revised the manuscript as following to address this issue. First, we generated the full-length antibody-receptor complex model using a second template, the recently published human α4β2 nicotinic receptor (PDB ID, 5KXI, 3.94 Å resolution). Although the overall structure of this model is similar to that built on the Torpedo AChR template, the alpha1 helix in the α4β2 nicotinic receptor has a “normal” conformation as seen in other monomeric and pentameric nicotinic receptor proteins and homologues. As a result, the modeled full-length antibody-receptor complex has an antibody-receptor binding interface that closely resembles the Fab35/nAChR α1 interface observed in our experimentally determined structure. These improvements increase the confidence of the modeling analyses. Second, we have eliminated any description of the membrane curvature hypothesis in the Results section and Abstract, and modified Figure 6 to focus on the possible effect of neighboring subunits on the antibody-receptor interaction (Figure 6) and the structural basis of antibody-mediated receptor cross-linking (Figure 6). Third, we have placed reference #4 in the correct context. Finally, we have limited our discussion of the hypothetical model of antibody cross-linking induced membrane curvature to two sentences in the Discussion section and converted Figure 7 into two supplemental figures to Figure 6 (Figure 6—figure supplement 1).

*3) The manuscript would have been strengthened by the addition of mutants and binding assays to specifically interrogate the interface seen in the X-ray structure. Although we appreciate that this would require considerable additional work, the authors can and should bring in the literature to help support the interface they see in the structure, as Luo et al., 2009 and others have predicted some of the interactions seen in the present study.*

We have added a section in the revised manuscript to discuss literature data supporting the interface interactions observed in the crystal structure. This can be found in the first two paragraphs of the Discussion section.

[Editors' note: further revisions were requested prior to acceptance, as described below.]

*The manuscript has been improved but there are some remaining issues that need to be addressed before acceptance, as outlined below:*

*1) There is something funny going in paragraph four of the Discussion section. A part of one sentence appears to have been deleted.*

Thank you very much for pointing it out. The sentence color seemed to become white during uploading process. We will make sure it will not happen again in this submission.

*2) It would be helpful for the authors to include a few remarks on the following two issues:*

*Although mAb 35 and mAb 198 are very similar in structure, other EAMG and MG antibodies that bind to this region may not be so similar or bind to identical epitopes. The high frequency of MIR antibodies in MG and EAMG sera was determined by competitive binding between mAbs with serum antibodies. Figure 1 clearly illustrates how large Fab 35 is with respect to the extracellular domain of α 1. Large intact mAb 35 is likely to compete with intact large serum autoantibodies for binding in this region to nearby but somewhat different epitopes than are recognized by mAb 35. mAB 192 is a similar case. The relative size of an antibody and an AChR is nicely illustrated in Figure 6. The variety of epitopes in the main immunogenic region is an important factor relevant to the Discussion, paragraph two, about trying to treat MG with mimotopes or for developing epitope-specific diagnostics.*

This is a very important point relevant to epitope-specific MG therapies and diagnostics. Before our crystal structure, it has been suggested that EAMG and MG antibodies may bind epitopes different from MIR and that these MG antibodies may be competed off by mAb35 through steric effect rather than direct epitope competition (Luo et al., 2009). Our structures (Figure 1 and Figure 6) indeed support this possibility, and we also mentioned that mAb192 may represent one such case (Discussion section, paragraph two). However, our structures also reveal that many MIR residues center at the antibody-receptor interface, and these MIR residues have been found to be important for the high affinity binding of a number of MG antibodies (e.g., mAb35, mAb210, and mAb132A). This is a rather surprising finding given the potential heterogeneity of nAChR antibodies mentioned above. Based on this observation, we suggest that molecular mimicries of the MIR and its immediate surrounding regions may be used to compete a large fraction of MIR-directed MG autoantibodies that share the same binding mechanism. We revised the manuscript to reflect both aspects of this point (Discussion section, paragraph two).

*In the Results section it is noted that the N terminal α helix of α 1 in Torpedo AChR differs from that in Bellone, Tang, Milius and Conti-Tronconi, 1989. It is worth noting that Morell et al., 2014 shows that variations in MIR structure have large effects on ACh sensitivity. AChR sensitivity involves both ACh binding sites and transmembrane regions. The transmembrane regions are greatly effected by lipid. Solubilization of Torpedo AChRs puts them in an inactive conformation that can be corrected by reconstitution into the proper lipid composition (C daCosta and J Baenziger (2013) Cell Structure 21: 1271-1283). Most crystallized receptors are solubilized and consequently contact lipids partially or completely removed. The 5-HT3 receptor after solubilization was crystallized bound to nano bodies directed at its C loops that function as potent antagonists (Hasssaine et al. (2014) Nature 512: 276-281). This too may have global conformation consequences. Torpedo AChRs are unique in having had their structures determined in their native membrane. Of course, artifacts in their structural determination might also occur.*

We appreciate the importance of the structural variations of the N-terminal α helix and the nearby MIR and are aware of the work described in reference 30. The suggestion that lipid may be a reason for the unique N-terminal α helix conformation observed in the Torpedo AChR is interesting. However, we feel that this is a complex question that is hard to resolve with information currently available. The packing interactions between the N-terminal helix and the MIR or MIR-equivalent region on various nAChRs and homologues are highly conserved. The strong chemical forces explain the highly conserved conformation between the N-terminal helix and the MIR or MIR-equivalent region on many published crystal structures of nAChRs and homologues. On the other hand, the Torpedo AChRs was solved by unique experimental approaches using native membrane. It is not entirely clear if the structural difference of the N-terminal helix is due to the membrane effect or intrinsic errors associated with different experimental techniques. Future studies may be needed to address this interesting and important questions. Because of this uncertainty, we modeled the full antibody-receptor complex using both the Torpedo AChrs and the a4b2 nAChRs to show that overall spatial arrangements of each components are similar regardless of the orientation of the N-terminal aloha helix. We have added one sentence to further clarify this point (paragraph two, subsection “Models of higher-order antibody-receptor structure”).